# Investigation of Peritectic Solidification Morphologies by Using the Binary Organic Model System TRIS-NPG

**DOI:** 10.3390/ma13040966

**Published:** 2020-02-21

**Authors:** Johann P. Mogeritsch, Mehran Abdi, Andreas Ludwig

**Affiliations:** Department of Metallurgy, Chair for Simulation and Modelling Metallurgical Processes, Montanuniversitaet Leoben, 8700 Leoben, Austria; mehran.abdi@unileoben.ac.at (M.A.); andreas.ludwig@unileoben.ac.at (A.L.)

**Keywords:** peritectic solidification, ESA, TRIS-NPG, Bridgman-furnace, MICRESS

## Abstract

Under pure diffusive growth conditions, layered peritectic solidification is possible. In reality, the competitive growth of the primary α-phase and the peritectic β-phase revealed some complex peritectic solidification morphologies due to thermo-solutal convection. The binary organic components Tris-(hydroxylmenthyl) aminomethane-(Neopentylglycol) were used as a model system for metal-like solidification. The transparency of the high-temperature non-faceted phases allows for the studying of the dynamic of the solid/liquid interface that lead to peritectic solidification morphologies. Investigations were carried out by using the Bridgman technic for process conditions where one or both phases solidify in a non-planar manner. Different growth conditions were observed, leeding to competitive peritectic growth morphologies. Additionally, the competitive growth was solved numerically to interpret the observed transparent solidification patterns.

## 1. Introduction

Many alloys of great industrial significance, such as steel, Al and Cu-alloys, and rare earth permanent magnets, show a phase diagram with a peritectic reaction. The characteristic of such phase diagrams is that, at peritectic temperature *T_p_*, the primary α-phase reacts upon cooling with the remaining liquid to transform into the peritectic β-phase (α + L→β). At this distinct temperature *T_p_*, a liquid of concentration *C_l_* is in equilibrium with an α-phase of concentration *C_α_* and a β-phase of concentration *C_β_*. Notwithstanding, alloys with compositions between *C_α_* and *C_β_* are called hypo-peritectic, those with compositions between *C_β_* and *C_l_* are called hyper-peritectic. Depending on the process conditions, a dendritic, cellular or planar interface can be obtained individually for each phase. 

Under dendritic/cellular growth conditions for both phases, the peritectic β-phase can solidify directly from the interstitial liquid instead of forming by the transition from α to β. The reason is the fact that the transformation kinetics and the rate of diffusion in the solid phase (α→β) is slower, in comparison to the growth kinetic and diffusion in the liquid phase (L→β). Below the limit of constitutional undercooling, where both phases are supposed to solidify in a planar manner, it is possible that neither the primary nor the peritectic β-phase can reach a growth state that corresponds to thermodynamic equilibrium. Under such conditions, alternative primary and peritectic layers may form perpendicular to the growth direction, as shown for the first time by Boettinger [1]. 

Advanced investigations with directional solidified peritectic alloys like Zn–Ag [2], Sn–Cd [3], Cu-Sn [4], Pb–Bi [5], Zn–Cu [6,7], Sn–Sb [8], Ti–Al [9], Fe–Ni [10], Ni–Al [11], YBCO [12], and Nd–Fe–B [13] show a variety of complex layered peritectic microstructures. The microstructures found in post mortem analyses of peritectic alloys are isothermal peritectic coupled growth (PCG), cellular peritectic coupled growth, discrete bands, island bands, and oscillatory tree-like structures. Trivedi suggested the first conceptual description of cyclic nucleation and growth under purely diffusive conditions [14]. 

Trivedi’s concept focused on the formation of peritectic banded structures, but only under pure diffusive growth. Hunzinger et al. [15] extended the concept by considering a criterion for nucleation. Thereby, an exact description is not possible today because of the following three main arguments. First, the presence of melt convection often changes the melt concentration locally as well as at the sample scale [16,17]. Second, the lateral spread of the phase that has just nucleated, competes with the forward growth of the present phase and thus incomplete bands, so-called islands or island banding, form [18,19]. Finally, the three dimensionality of an unsteady solid/liquid (s/l) interface motion may lead to a dynamic phase interaction where nucleation is of less importance [17].

It is especially the case that all findings are based on post mortem analyses of metal samples, which has drawn attention by the authors to use the transparent, organic non-faceted/non-faceted (nf/nf) components TRIS (Tris-(hydroxylmenthyl) aminomethane) and NPG (Neopentylglycol) [20] as a model system for peritectic solidification [21,22,23,24,25,26]. Thus, it became possible to observe, in situ, the mechanisms that lead to bands, island bands, and isothermal PCG [21]. 

In previous studies, the growth velocities were shown to be, for both phases, solidified as cells and/or dendrites with the transition from the primary phase to the peritectic β-phase, because the preferred phase revealed some spectacular observations of compact seaweed type growth [22]. For cellular/dendritic growth conditions, observations indicated that the primary α-phase grew continuously in a cellular/dendritic pattern, as expected. In contrast, the peritectic β-phase solidified dendritically and discontinuously within the intercellular/dendritic liquid [23] or both phases solidify in a dendritic/cellular manner in the form of an oscillating coupled growth [24]. For process conditions that had both phases solidified in a planar, the main findings were that a layered peritectic solidification pattern, in the form of island bands, later transform into unstable PCG or nucleation events of the β-phase at the s/l interface and were directly overgrown. Accompanying studies with seeding particles as tracers have shown that convection in the melt is due to buoyancy and due to plumes that formed by the migration of residual melt inclusions in the solid [25]. Additionally, numerical investigations [26] were performed to estimate the missing physical data of the model system TRIS-NPG. 

Unfortunately, in such in situ observations, the similar optical appearance of the two transparent phases made the exact interpretation of the optical investigations difficult. The only option to distinguish between the two phases is their different growth dynamics close to the constitutional undercooling. Changes in the growth morphologies can be taken as an unmistakable hint of the transition from the primary to the peritectic phase growth and to identify the two different phases. 

The aim of this article is to investigate nucleation events and competitive growth morphologies under process conditions with one or both phases solidified in a non-planar way. In order to do so, we have varied the alloy concentration from *x* = 0.47 mol% to *x* = 0.54 mol fraction NPG and the pulling rate from *V_p_* = 0.1 μm/s to *V_p_* = 0.32 μm/s. The temperature gradient was estimated to be *G_T_* = 6.65 K/mm. Additionally, accompanying numerical investigations was the phase field based software MICRESS, which was done to support the interpretation of the observed solidification morphologies.

## 2. Materials and Methods 

The transparent organic peritectic system TRIS-NPG, see Figure 1a, show orientationally disordered crystals C_l_ and C_F_, called non faceted or plastic phases, which solidify in a metal-like structure. Therefore, the compounds can be used as a model system for peritectic solidification morphology. Plastic phases are contrast due to the low temperature facetted phases being transparent and colorless. Figure 1b shows the peritectic plateau of the TRIS-NPG phase diagram in detail. For the purposes of generalization, the C_l_ and C_F_ phases were renamed within this paper by α or primary phase and β or peritectic phase. Furthermore, it should be noted that a phase diagram represents the state of equilibrium, whereas peritectic layered solidification morphologies are in a non-equilibrium state. 

Layered structures are predicted for a solidification speed *V* below critical velocity *V_c_* [1,14]. At the limit of the constitutional undercooling, for *V* ≤ *V_c_*, a stable planar front grows at the corresponding solidus temperature *T_s_* of the phase. Otherwise, if *V* > *V_c_*, the planar interface becomes unstable and transforms to cells and/or dendrites to reduce the zone of constitutional undercooling. The critical solidification velocity *V_c_* can be expressed by:(1)$$V_{C}=\frac{D_{L}\cdot G_{T}}{T_{l}-T_{s}}.$$

Here, *D_l_* is the diffusion coefficient in the liquid, *G_T_* the temperature gradient within the adiabatic gap, *T_L_* the liquidus temperature, and *T_s_* the solidus temperature. In the peritectic region the values of the two temperature differences, *ΔT_α_* (*T_l_* − *T_s_*) and *ΔT_β_*, are dissimilar (see Figure 1b), particularly when considering the elongated slopes within the metastable region. Therefore, for a definite pulling velocity *V_p_*, it is possible that one phase grows below the limit of constitutional undercooling and the other one grows above the limit of the constitutional undercooling. Additionally, the shift of concentration in the liquid ahead of the s/l interface has an impact on *ΔT_α/β_* and thus on the critical velocity *V_c_*.

The organic compounds were delivered with a purification of 99% for NPG and 99+% for TRIS. A drying process to reduce the water content increased the purity of NPG and TRIS was used without additional purification. Both compounds had to be handled in a protecting atmosphere. Compounds were prepared by mixing the substances in a solid state, fusing them together and homogenizing them by melting and cooling, respectively. Since the molten substances had high and different steam pressures, preparation took place in small hermetically sealed container to avoid uncontrollable concentration change by evaporation. The obtained compounds with an accuracy of ±0.0002 mol fraction NPG were ground to powder before further use. More details on purification, alloying and filling are given in Reference [21,22]. 

The observation of the dynamics of the s/l interface was carried out by using the Bridgman technique, see Figure 2. The furnace consisted of two brass blocks separated by a gap fixed by ceramic covers for thermal insulation. The brass blocks and the ceramic cover are separable to allow a simple sample change. In order to fix and guide the sample, a slot of 2.5 × 0.4 mm was additionally milled into the brass blocks. The upper brass block, with a sample contact zone of 10 mm, served as a hot zone and the lower one with a sample contact zone of 40 mm as a cold zone, with a 7 mm gap as an adiabatic zone. The temperature inside the brass parts was controlled by electrically resistant heaters and measured with Pt-100 temperature sensors mounted in each brass block. The sample was pulled vertically (*V_p_*) at a constant PC-controlled velocity through the temperature gradient (*G_T_* = 6.5·10^−3^ ± 0.2·10^−3^ K/m) within the adiabatic zone. The cooling rate $\dot{T}$ is given by:(2)$$\dot{T}=V_{p}\cdot G_{T}$$

The pull rate was adopted for each experiment to change the process conditions, but the temperature gradient was maintained.

The observation of the dynamic of the solid–liquid interface morphology was carried out with a ZEISS microscope equipped with a CCD camera. During the solidification experiments, images of the interfacial morphology were taken for later evaluation and the corresponding temperatures of the two brass parts were recorded. The coordinate system used in this paper is displayed within the detail of the sample in Figure 2. 

The sample consists of a rectangular glass tube illuminated by glass windows placed in the ceramic covers within the adiabatic zone. The sample was manufactured by capillary force filling the rectangular quartz tubes (100 × 2000 μm cross sectional area, 100 μm glass wall thickness) with the organic compound and sealed afterwards with a UV-hardening glue. The sealed glass sample was placed into the furnace and remained stationary for 1 h to reach a state of thermal equilibrium before the solidification experiment was carried out. In this paper, the statements on the left (*X* ≥ 1500 µm), middle (500 ≤ *X* ≥ 1500 µm) and right (*X* ≤ 500 µm) refer to the morphology close to the glass sample’s front side (*Y* = 0 µm), viewed from the direction of the CCD camera. In the case of the spatial depth of the sample, the terms front, (*Y* = 0 µm), center (*Y* ≈ 50 µm), and back (*Y* ≈ 100 µm) are used.

Numerical investigations were realized with the phase field based commercial software program MICRESS. The object of the simulation was to support the interpretation of the observed solidification morphology and to obtain additional information on the concentration distributions during the solidification process. The cell size of the simulation domain was selected by 4 × 4 × 4 µm^3^ with 50 cells in the *X*-direction (200 µm), 25 cells in *Y*-direction (100 µm), and 800 cells in *Z*-direction (3200 µm), which gave a total of 1,000,000 cells. The selected cell size and number of cells still allows for an acceptable simulation time and to be in the position to make a quantitative statement. Within the domain, a 50 µm-high single grain of the primary α-phase was set at the bottom, separated from the domain wall by a narrow melt film (1 µm) according to the experimental observation. The boundary conditions at the domain walls were set as “isolated” for the X-Z plane to represent the front and rear glass walls, “periodically” in the Y-Z plane to represent an infinite sample width (real 2000 μm, simulation 200 μm), “isolated” at the X-Y bottom plane and a “fixed” concentration at the X-Y top plane. The size of the domain only corresponds to a part of the total sample length inside the adiabatic gap. It covers the molten part and enables the observation of the temperature range down to the solidification temperature of pure NPG. During the simulation time, the “moving_frame” option in MICRESS enabled the domain to follow the solidification front. There are two nucleation models available in MICRESS, “seed_density” or “seed_undercooling”. “Seed_density” described heterogeneous nucleation in the melt, but “seed_undercooling” enables a nucleation event at the s/l interface. Hence, the “seed-undercooling” model was selected. The process conditions of the Bridgman furnace were represented by a constant cooling rate $\dot{T}$ and a constant temperature gradient *G_T_*. The corresponding physical and numerical parameters, which were used for MICRESS, are given in Table 1.

The distribution coefficient and the slopes of the α- and the β-phase at the peritectic temperature, *T_p_*, were taken from Figure 1b. The α solidus slope was elongated within the metastable region (*x* > 0.47) with a nucleation for concentrations, even when pure NPG at 400 K occurs. 

## 3. Results

### 3.1. Initial Morphologies after One Hour in Rest

After placing the sample into the preheated Bridgman-furnace, it was kept for one hour in rest. Thus, part of the organic compound melts. Within the plastic phase, liquid inclusions were visible in the form of drops (black dots), see Figure 3. The size of the droplets was approximately 80 ± 10 µm. This indicated that the droplets filled out almost the entire sample depth. According to the phase diagram the droplets consisted of highly NPG-enriched organic compounds. The solid transparent plastic phase showed a polycrystalline structure consisting of individual grains imbedded in a liquid film. Thereby, either there was only one phase or both phases were recognizable from the beginning. In the case where both phases were present, regions of different grain structures were noticeable, as shown in Figure 3a—the so called type A in this paper, or a more or less sharp horizontal line within the solid like structure in Figure 3b, named type B. According to the phase diagram, the structure or region that appeared at a higher temperature level is the primary α phase. 

For type A (Figure 3a, hypo-peritectic concentration *x* = 0.473), two phases were formed parallel to the temperature gradient. This was visible in form of different polycrystalline structures. In the middle (*X* ≈ 500–1500 µm), the morphology showed roundish grains, whereas the grains close to the sample side walls (1500 > *X* < 500 µm) were elongated. It has to be mentioned that the s/l boundary showed a curved interface down to a lower temperature level close to the side walls. As discussed in Reference [23], it is quite likely that the isotherms within the samples are not curved, especially if the sample is in rest. According to the statement above, this suggested that the central region with an s/l interface at a higher temperature level was formed by the α-phase, whereas the side region at a lower temperature level consisted of the β-phase. 

More common was the morphology for type B, as shown for the hyper-peritectic concentration *x* = 0.537 in Figure 3b. Two optically distinct structures, orthogonal to the temperature gradient, were visible in the solid. A 130 µm thick layer in direct contact with the melt and the remaining solid. Both plastic phases showed a polycrystalline structure of grains coated by liquid films. Due to the transparency of the plastic phases, it can be seen that most of the grains continue in the entire depth of the sample. Additionally, in contrast to the s/l interface of type A in Figure 3a, the s/l boundary is straight or smoothly curved. With the help of the phase diagram, the structure closest to the melt can be identified as the primary α-phase and the other as the peritectic β-phase. 

The results of the direct solidification experiment showed that (i) the primary phase grew exclusively and no nucleation event happened. This was observed for more than 80% of the experiment. Otherwise, (ii) both phases were presented in their initial form of type A, or mostly type B, and competitive growth led to several peritectic morphologies. In all cases, a peritectic reaction was not detected. Statement (i) was not further investigated due to the missing growth or nucleation of the peritectic phase. Results from statement (ii) are presented in the following sections, and are compared and evaluated with the findings of the numerical investigations. 

### 3.2. Competitive Growth Morphologies Caused by Nucleation Events of the Peritectic Phase

Nucleation events were observed for concentrations in the hypo-peritectic region at the onset of the peritectic plateau (*x_α_* = 0.47), near the peritectic concentration (*x_p_* = 0.515), and in the hyper-peritectic region near the final point of the plateau (*x_L_* = 0.54). 

For the hypo-peritectic concentration *x* = 0.473 and a *G_T_*/*V_p_* ratio of 2.5∙10^10^ K∙s/m^2^ ($\dot{T}$ = 1.7 × 10^−3^ K/s), a polycrystalline structure was visible. During the melting process, the liquid film at the grain boundaries migrated due to melting and solidification, known as the temperature gradient zone melting TGZM [25,28]. After one hour, the liquid films nearly disappeared. Initially, both phases were present in the form of type A, as shown in Figure 3a. The primary phase grew against the sample pulling direction, but the growth of the s/l interfaces was slower than the sample movement. This was indicated by a change in the position of the s/l interface within the adiabatic gap from the hot zone towards the cold zone. This can be explained by the fact that the s/l interface tries to move from the liquidus temperature level to the solidus temperature level. However, solidification was preferable in the area of the sample center. As a result, the s/l interface curved towards the hot zone. Additionally, within the first 2460 s, the solidification structure changed from a banded planar front to dendritic growth. Finally, only the primary phase was recognizable.

After *t* = 22.020 s, the β-phase nucleated at a temperature range of *T* = 389 ± 2 K on the sample left side wall and grew along the interstitial liquid (Figure 4a). After a time-delay of *Δt* = 4.140 s, a further nucleation event happened, located on the opposite sample side wall. The newly formed β-phase spread horizontally from both nucleation regions toward the middle of the sample. Thereby, the initial phase grew dendritical and the nucleated phase was more cellular. The distance between the dendrite tips and the cell tips remained constant with *Δx* = 370 µm and corresponded to the temperature gradient at a temperature difference of *ΔT* = 2.4 ± 0.1 K (Figure 4b).

A nucleation event was also detected for the hypo-peritectic concentration with *x* = 0.502, close to the peritectic concentration (*x* = 0.515), and an *G_T_*/*V_p_* ratio of 4.1∙10^10^ K∙s/m^2^ ($\dot{T}$ = 1.0∙10^−3^ K/s). As described in the section above, initially, two optically and clearly distinguishable phases were visible and were definitely of type B, which was similar to Figure 3b. One 230–390 µm thick solid layer, the primary phase, interspersed with almost parallel vertical liquid films in direct contact with the melt and the remaining solid, the peritectic phase with a polycrystalline structure, was embedded in liquid films. Since both phases are transparent, it was not possible to determine whether the structure of the peritectic polycrystalline morphology continued throughout the depth of the sample. At the beginning of the solidification investigations, a curved boundary surface, consisting of several cells, was formed. Simultaneously, the s/s interface between the primary phase and the peritectic phase strictly followed the movement of the glass sample until the boundary line was out of the observation window. This was an indication that no peritectic transformation happened. At *t* = 35.460 s, nucleation of the peritectic phase occurred in the temperature range *T* = 392 ± 2 K in the area of the left side wall of the sample, see Figure 5a. The peritectic phase spreads close to the cell root at *T* = 405 ± 1 K, horizontally from the left side up to the middle of the sample. Thereby, the formation of heart-shaped structures was observed during the horizontal spread, as shown in Figure 5b.

For a hyper-peritectic concentration of *x* = 0.537 (peritectic plateau: 0.47 ≤ *x* ≤ 0.54) and type A with *G_T_*/*V_p_* = 2.3∙10^10^ K∙s/m^2^ ($\dot{T}$ = 1.9·10^−3^ K/s), the solidification morphology changed from the initial planar s/l interface via cellular growth to a dendritic solidification morphology within the first *Δt* = 3.600 s. At *t* = 8.850 s, nucleation took place in the temperature range of *T* = 398.7 ± 1.0 K. The propagation of the β-phase happened as a compact seaweed type. The entire primary phase was overgrown within *Δt* = 90 s, see Figure 6a. As a consequence, only the peritectic phase was in the position to grow further on. It has to be remarked, that the solidification morphology changed from a more dendritically structure of the primary phase to cellular for the peritectic phase (Figure 6b). Similary solidification patterns were observed with a concentration of *x* = 0.54 and *G_T_*/*V_p_* = 2.0∙10^10^ K∙s/m^2^ ($\dot{T}$ = 2.1·10^−3^ K/s) [22].

### 3.3. Competetiv Growth of the Primary and Peritectic Phase without Nucleation

For concentrations in the hypo-peritectic region close to the peritectic concentration, band-like growth structures were also observed, which are not caused by a nucleation event. Since all experimental results showed approximately the same progress, only the obtained result of sample *x* = 0.504 (*G_T_*/*V_p_* = 2.8∙10^10^ K∙s/m^2^ or $\dot{T}$ = 1.5∙10^−3^ K/s) is discussed in detail as a representative for the similar solidification morphologies observed for the concentrations *x* = 0.515 and *x* = 0.511 (both at *G_T_*/*V_p_* = 2.5∙10^10^ K∙s/m^2^ or $\dot{T}$ = 1.7∙10^−3^ K/s). 

As displaced in Figure 7a, the non-melted area showed a polycrystalline structure, traversed by inclusions and one gas bubble, as well as an approx. 260–280 µm thick α-phase layer. The region on the left side of the sample showed different grain structures. This indicated an overlapping of both phases in the depth of the sample. The flat cellular s/l interface changed to a cellular morphology within *Δt* = 4740 s. In contrast to the processes described in the previous chapter, no nucleation event happens. Instead, the peritectic phase grows from a small region at the s/s boundary along the existing liquid films within the primary phase up to the cell root (Figure 7b).

From the moment the peritectic phase reached the interstitial liquid at the sample wall, it grew in a horizontal direction. As soon as the β-phase reached over the entire sample length, the new preferred growth direction was vertical. In the mutual growth struggle, the peritectic phase enveloped the primary phase in the middle and overgrew it in the side regions (Figure 8a). As time went on, however, the solidification morphologies of both phases changed. The α-phase changed from a cellular to dendritically structure and the flat s/l interface of the peritectic phase grew cellular-like (Figure 8b). As in Figure 4b, the distance between the dendrite tips and the cell tips remained constant at a temperature difference of *ΔT* = 2.4 ± 0.1 K.

### 3.4. Numerical Interpretation of the Observed Solidification Morphologies

The numerical investigations were carried out for the concentrations *x* = 0.48, 0.49, 0.50, 0.51 and 0.52 with *G_T_*/*V_p_* = 4.2, 2.8 and 2.5·10^10^ K∙s/m^2^ ($\dot{T}$ = 1.0, 1.5 and 1.7·10^−3^ K/s) for the selected process conditions. The results obtained show similar solidification morphologies, different only in the actual point in time where nucleation happened—a consequence of the different selected cooling rates. For further evaluation of the experimental data in this paper, the description of only one numerical result might be sufficient. Therefore, we present and analyze the results for the concentration *x* = 0.52 and the *G_T_*/*V_p_* ratio of 2.5·10^10^ K∙s/m^2^ ($\dot{T}$ = 1.7∙10^−3^ K/s). Originally, MICRESS displayed the calculated data in two-dimensional images, layer by layer (2 µm distance). To improve the optical representation, the numerical results were converted into a three-dimensional graphic using the software PARAVIEW.

Figure 9a shows the initial planar s/l interface of the primary α-phase, blue colored, and the liquid in half transparent yellow. The three-dimensional graphic allows for the display of the solidification morphology and to detect the progress of solidification within the sample. According to the experimental results, the α-phase solidified dendritically, as shown in Figure 9b. Looking more closely at the surface of the α-phase shown in Figure 9a, it can be seen that a thin film of liquid remains between the front and back glass walls and the solid. It should be noted that, due to the selected domain boundary conditions, only part of the total sample width was displayed. Therefore, no liquid film was present on the side walls due to the periodic boundary conditions.

As soon as a β nucleation event in the numerically predefined temperature range of 397 K took place, the peritectic phase started to grow along the existing liquid film between the glass walls and the primary phase (Figure 10a). The growth on both inner sides of the glass wall envelops the primary phase (Figure 10b). 

## 4. Discussion

The evaluation of the experiments was carried out by (i) comparing the temperature level over the experimental period, (ii) the time of the nucleation event and (iii) the spatial interpretation of the dynamic of the s/l interface by using numerical simulation. 

(i) The s/l interface temperature could be determined due to the position within the adiabatic gap. It should be noted that the interface or the imaginary enveloping line of cells or dendrites was curved against the hot zone, as discussed in Reference [22]. Therefore, the closest to the hot zone α/l interface was used as the reference value for all experiments. Figure 11a shows that the movement of the s/l interface from the liquidus temperature to the solids temperature needs approximately 3 h (10.800 s). Afterwards the interfaces grew at a temperature level of 408 ± 2 K. This temperature interval covers a concentration range of 0.55 ≤ x ≤ 0.80 mol fraction NPG, assuming there is sufficient accuracy of the phase diagram. 

The time-dependent temperature at the β/l and α/β interface is shown in Figure 11b. If the peritectic β-phase was present at the beginning of the experiment; the s/s interface follows the pulling rate of the sample until it disappears from the observation area. These are recognizable as steeply sloping dots in Figure 11b. As soon as the nucleation events happens, the β-phase grew at a temperature level of about 405 ± 1 K. Thereby, the observed nucleation events happened close to the side glass walls—see Reference [21]. This is understandable since the α/l interface was toward the side region curved and the liquid there was NPG-enriched. In that case, no nucleation event occurred and the peritectic phase grew through the primary phase; the s/l interface temperature also dropped down to the temperature level of 405 ± 1 K within the first 3 to 4 h. This means, according to the phase diagram, that the peritectic phase grew with a concentration range of 0.75 ≤ x ≤ 0.85 mol fraction NPG, as shown in Reference [22]. 

It should be noted that the sample *x* = 0.54 was the only concentration where the peritectic phase, in the form of a compact seaweed type, overgrew the primary phase. In all other cases, the primary phase was preferred. As a remark, the concentration *x* = 0.54 is located at the end of the peritectic plateau. According to the phase diagram for slightly higher NPG concentrations, the peritectic phase should grow solely. 

The numerical investigation, based on Reference [25], was used to investigate the concentration distribution within the interstitial liquid and to interpret the observed spatial solidification morphologies. It shows that the maximum concentration increases within the first 4 h (14.400 s) for all performed simulations to nearly 90 wt.% NPG, which corresponds to a *x* = 0.94 mol fraction NPG and stayed constant afterward, see Figure 12a. The corresponding concentration gradient within the liquid at the moment of the nucleation event (*t* = 12.720 s) is shown in Figure 12b. The given numerical boundary conditions enable a nucleation event according to the experimental findings at 400 K or below. As a conclusion, nucleation takes place in an almost pure NPG liquid.

Further on, the observed growth structure of the peritectic β-phase, as shown in Figure 13a, was investigated. The observed horizontal growth direction of the peritectic phase always took place at the same level close to the primary structure roots. Additionally, during the horizontal propagation of the peritectic phase, the transparent property of both phases allowed for the observation of recurring patterns in the form of patches. The structure and brightness of the boundary lines in the experimental images suggested that the peritectic β-phase grows at different depths. In Figure 13b–d, the numerical study shows that, during the solidification, a liquid film is always present between the glass walls and the α-phase. As soon as the necessary preconditions for the nucleation event are met, the peritectic phase grows along the liquid film between the glass wall and the α solid. When β-growth reaches the interstitial melt in the gap between the primary structure and the glass wall, it grows in the horizontal direction, as well as through the interstitial melt to the opposite glass wall. This causes the formation of β-patches, as shown in Figure 13b. Here, the peritectic phase spread in both directions and enveloped the primary phase (Figure 13b,c). The same effect happened when, in some experiments, the peritectic phase grew through the liquid films. A local favorable concentration distribution in the liquid film enables the growth of the peritectic phase through the existing α-phase, like in Figure 7b. 

## 5. Conclusions

The dynamics of the s/l interface for process conditions where one or both phases solidify in a non-planar manner was investigated. Different peritectic solidification patterns were found for concentrations within the entire peritectic plateau. By using the Bridgman-technique, the s/l interfaces showed, after one hour in rest, a slightly curved boundary and the solid a polycrystalline morphology. In the case where both phases were present, two different structures were discernible, (i) both phases existed in different regions and in contact with the melt pool ahead the s/l interface, called type A in this paper. The peritectic phase was situated close to the side walls and the primary phase was centered. Additionally, the s/l interface was curved in such a way toward the side walls to comply with the thermodynamic boundary conditions of different melting points. (ii) Only the primary α-phase was in direct contact to the melt pool in the form of a small layer in front of the peritectic β-phase, named type B by the authors. 

The competitive growth was triggered by two events. Either a nucleation event of the β-phase within the interstitial liquid or the growth of the initial present peritectic β phase through the existing liquid films between the primary α-phase. For both cases, the peritectic phase spreads horizontally and envelops the primary phase at a certain distance to the phase tips. This is possible because, during the growth of the primary phase, a melt film existed between the front glass plate and the growing primary phase. 

The dynamics of the s/l interface showed, during the horizontal growth, the formation of patches. Numerical investigations show that this is a result of the growth of the peritectic phase from one side of the glass wall through the interstitial liquid to the opposite glass wall. After complete propagation in the horizontal direction, the peritectic phase enveloped the primary phase. In the subsequently competing growth, the primary phase remains the preferred phase. In contrast, the nucleation events for concentrations x ≅ 0.54 led to a solidification pattern in the form of compact β-seaweed, whereby the primary α-phase was overgrown by the peritectic β-phase due to the equivalent thermodynamic option between both phases. 

## Figures and Tables

**Figure 1 materials-13-00966-f001:**
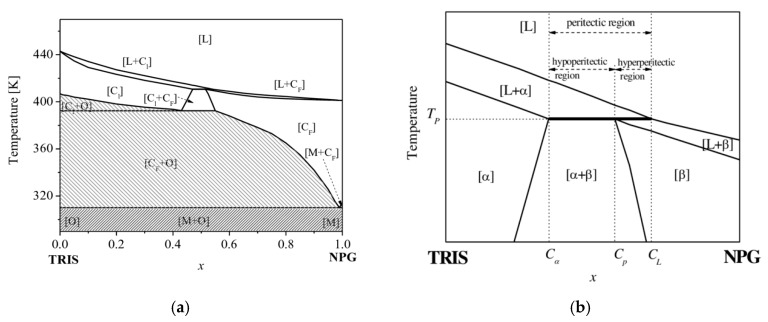
(**a**) TRIS-NPG phase diagram redrawn after Barrio [23]. In the dark shaded region, both low temperature substances [M + O] are faceted and not miscible. (**b**) Details of the peritectic region with the transparent non-faceted high temperature substances [C_l_ + C_F_], renamed as α and β phases [27].

**Figure 2 materials-13-00966-f002:**
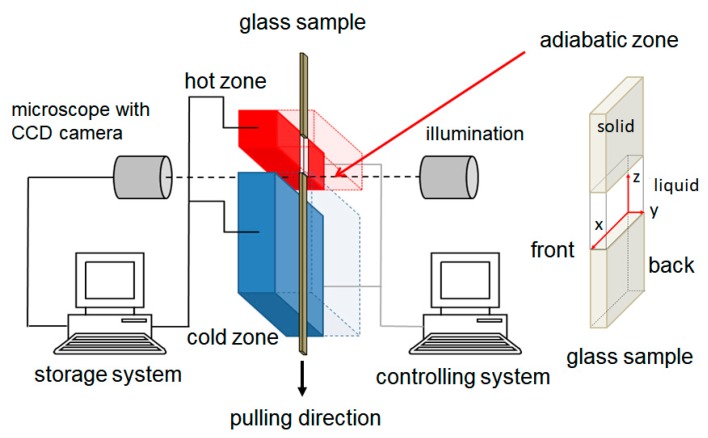
Sketch of the experimental set-up. The ceramic shelter to keep the brass plates in shape and act for thermal insulation is not shown. Next to the sketch, detail of the sample and coordinate system within the adiabatic zone.

**Figure 3 materials-13-00966-f003:**
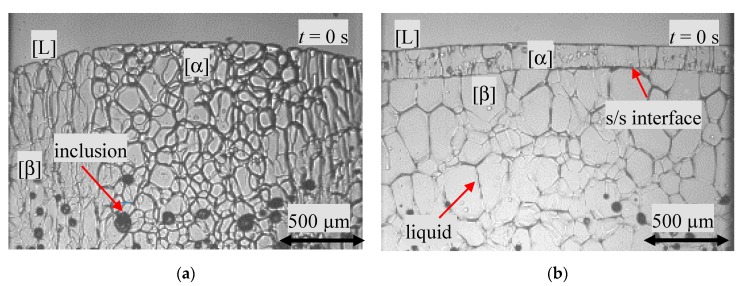
(**a**) Initial morphology of type A for a hypo-peritectic sample with *x* = 0.473 ± 0.0002 and (**b**) type B for a hyper-peritectic sample with *x* = 0.537 ± 0.0002 after 1 h in rest within the Bridgman-furnace. The black dots in both images are NPG-enriched liquid inclusions. In (**b**), two solid phases, seperated by a horizontal line, are recognizable. The images show a width of 2.000 µm.

**Figure 4 materials-13-00966-f004:**
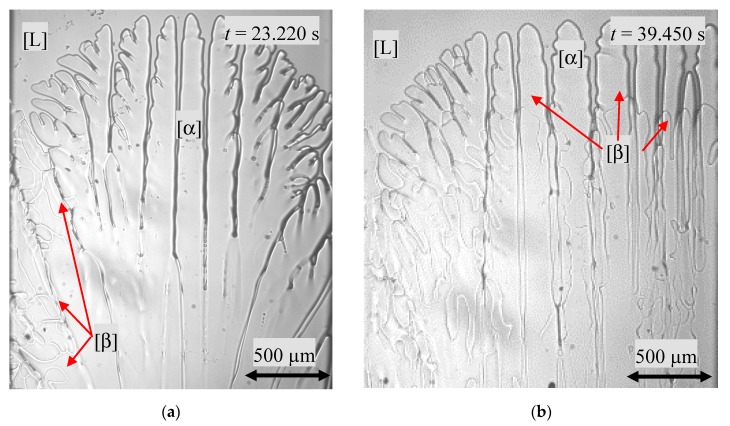
Solidification morphology for a hypo-peritectic sample with *x* = 0.473 (**a**) nucleation event at the left side of the image and (**b**) simultaneous growth of both phases in a dendritic/cellular manner. The images show a width of 2.000 µm.

**Figure 5 materials-13-00966-f005:**
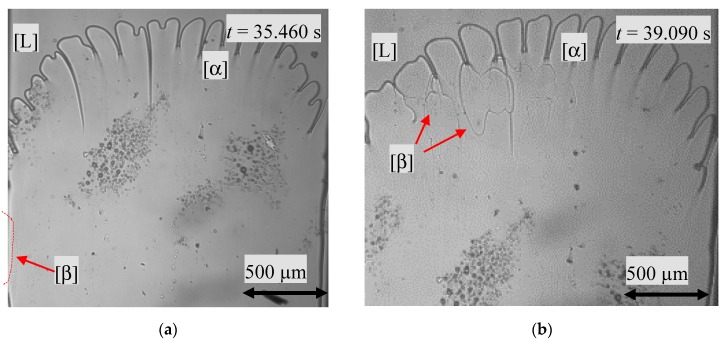
(**a**) nucleation event of the β-phase indicated by a red line and (**b**) dispersal of the peritectic phase along the interstitial liquid of the primary phase. The images show a width of 2.000 µm.

**Figure 6 materials-13-00966-f006:**
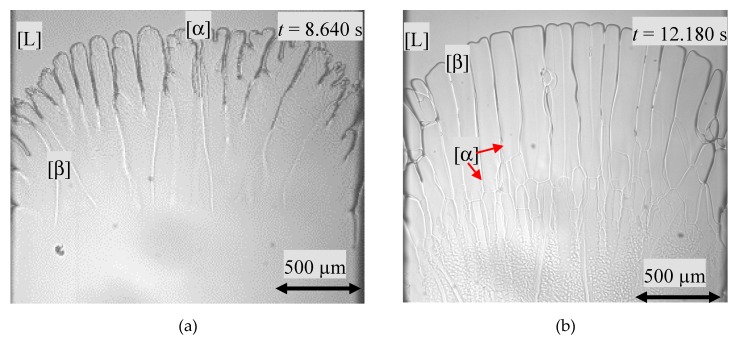
(**a**). Spread of the peritectic phase as a compact seaweed type. The peritectic phase overgrew the primary phase. (**b**) Only the β-phase grew and the leftover of the α-phase dendrites can be seen as fine lines. Additionally, the compact seaweed type of the β-phase, in the form of cauntless dots, is shown. The images show a 2000 µm width.

**Figure 7 materials-13-00966-f007:**
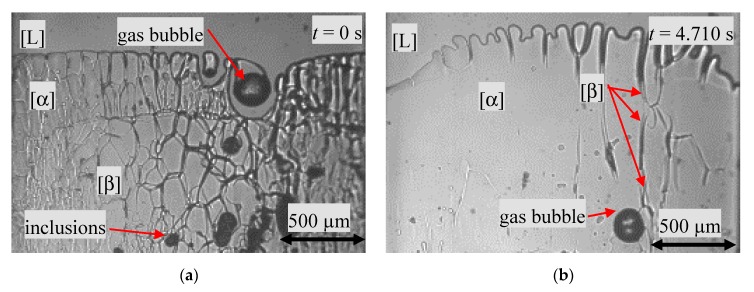
(**a**) The image of the sample shows the melt, two different solid phases, separated by a horizontal line, a gas bubble and various inclusions of NPG enrich melt. (**b**) Growth of the peritectic phase through small liquid channels across the α-phase toward the bulk liquid.

**Figure 8 materials-13-00966-f008:**
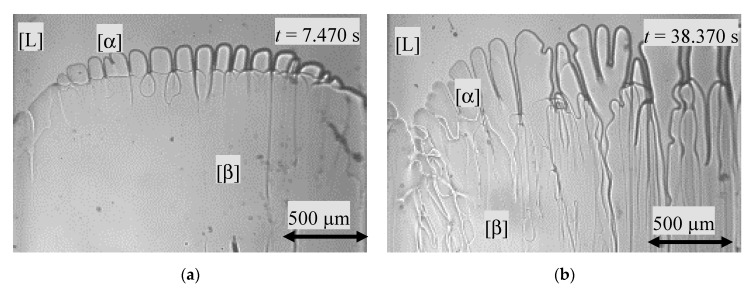
(**a**) The peritectic phase has enveloped the primary phase, but not entirely overgrown it. The solidification morphology of the primary phase shows deep cells from the peritectic phase flat cells. (**b**) The growth morphology has shifted from a cellular to a dendritic-like growth morphology.

**Figure 9 materials-13-00966-f009:**
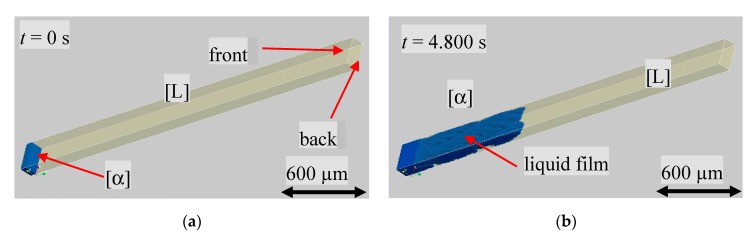
Three-dimensional representation of the α-phase (blue). The inner volume of the glass sample is delimited by white lines and the melt is shown in a semi-transparent yellow. The distance between the front and back corresponds to the 100 µm inner depth of the glass sample. (**a**) Initial planar s/l interface, (**b**) dendritically solidification morphology shortly before a β nucleation event took place.

**Figure 10 materials-13-00966-f010:**
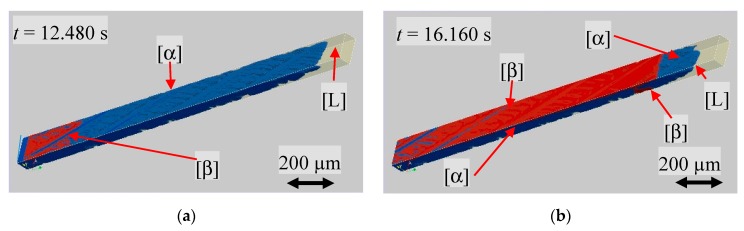
(**a**) Nucleation event of the peritectic β-phase (red) at the numerically set temperature of 397 K and (**b**) growth of the β-phase along both glass walls.

**Figure 11 materials-13-00966-f011:**
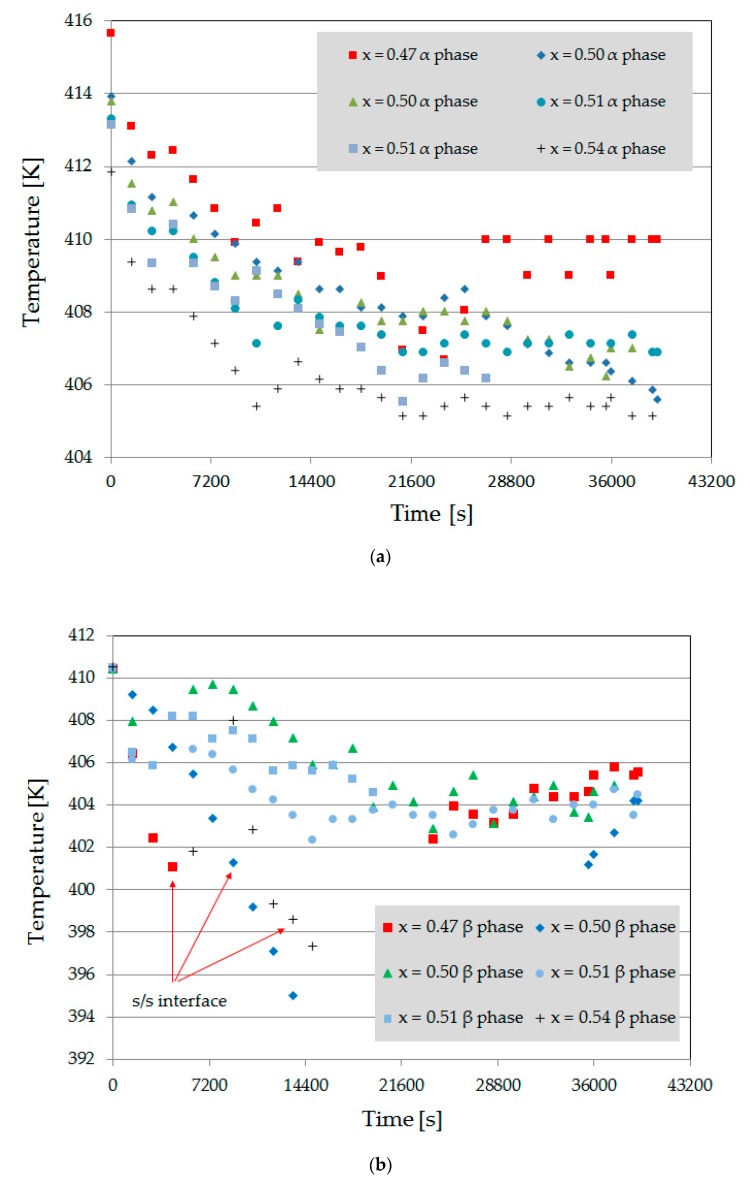
(**a**) Time dependent temperature level of the s/l interface for the primary α-phase and (**b**) the s/l and s/s interfaces of the peritectic β-phase.

**Figure 12 materials-13-00966-f012:**
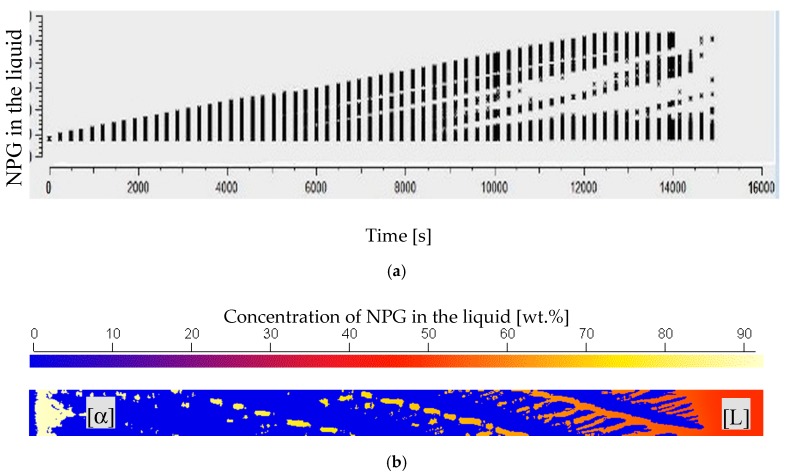
(**a**) Time-dependent evolution of the max. Concentration in the interstitial melt up to 14.400 s and (**b**) concentration distribution along the glass wall just before β nucleation happens at *t* = 12.720 s for an alloy with *x* = 0.52.

**Figure 13 materials-13-00966-f013:**
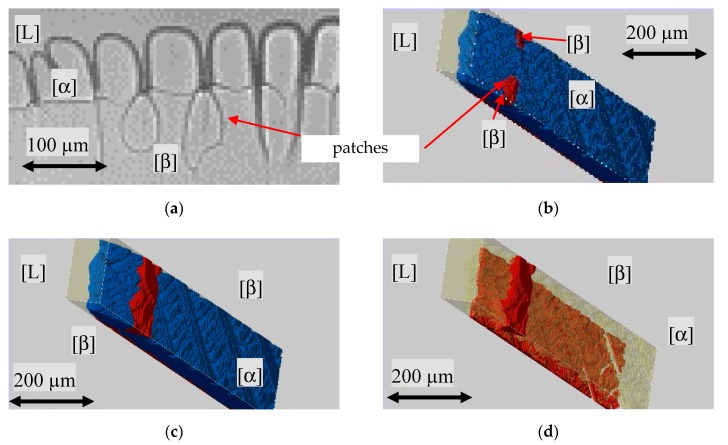
(**a**) The peritectic β-phase appears in the transparent model system with an almost flat s/l interface and some patches. (**b**,**c**) show that the primary α-phase (blue) is enveloped by the peritectic β-phase (red). (**d**) The coating takes place by the peritectic β-phase (red) growing through the interstitial melt (transparent light yellow) of the primary phase (transparent).

**Table 1 materials-13-00966-t001:** Numerical and physical parameter [26].

Names	Symbol	Unit	Value
surface energy, α-phase	*σ_s/l_* _,*α*_	J/cm^2^	1.0∙10^−6^
surface energy, β-phase	*σ_s/l_* _,*β*_	J/cm^2^	5.0∙10^−5^
cooling rate	$\dot{T}$	K/s	1.76∙10^−3^
entropy, α-phase	*S_f_* _,*α*_	J/cm^3^∙K	0.5
entropy, β-phase	*S_f_* _,*β*_	J/cm^3^∙K	0.8
interfacial stiffness coefficient	*σ**(*Θ*)	-	0.3
interfacial mobility coefficient, α-phase	*μ(Θ)_α_*	-	0.02
interfacial mobility coefficient, β-phase	*μ(Θ)_β_*	-	0.01
temperature gradient	*G_T_*	K/cm	65

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
