# Peer review of "Investigation of Peritectic Solidification Morphologies by Using the Binary Organic Model System TRIS-NPG"

_materials, 2020, doi:10.3390/ma13040966_

Round 1

Reviewer 1 Report

Dear Authors,

Your article concerns the use of TRIS-NPG organic components to predict the course of peritectic transformation. The article is characterized by good structure and composition, and its content is assessed as interesting and worth publishing. While reading the article, I had a few comments that, hopefully, will help shape the final version of your article.

In the abstract please explain the abbreviation: TRIS-NPG (as in line 57)

Line 58: too many references cited in one parenthesis.

Line 66: reference [24] should be cited earlier, in line 58.

Figure 1 caption: remove: “taken from”

Lines 122, 139, 149: remove power by unit.

Figure 2: I suggest describing both schemes as a) and b)

Table 1: add a row with column names: name, symbol, unit, value.

References: format according to the journal’s guidelines. Most of the cited papers have been published quite a long time ago. Almost half of the items in the list are the works of the authors. By default, it is assumed that the share of authors' own work should not exceed 25-30%. I suggest removing less important items and citing several recent (last 2 years) articles from literature. Maybe the following jobs will be useful:

Luo, S., Liu, G., Wang, P., Wang, X., Wang, W., & Zhu, M. (2020). Metallurgical and Materials Transactions A, 51(2), 767-777.

Nagasivamuni, B., Wang, G., StJohn, D. H., & Dargusch, M. S. (2019). Effect of ultrasonic treatment on the alloying and grain refinement efficiency of a Mg–Zr master alloy added to magnesium at hypo-and hyper-peritectic compositions. Journal of Crystal Growth, 512, 20-32.

Abraham, S., Bodnar, R., Lonnqvist, J., Shahbazian, F., Lagerstedt, A., & Andersson, M. (2019). Investigation of Peritectic Behavior of Steel Using a Thermal Analysis Technique. Metallurgical and Materials Transactions A, 50(5), 2259-2271.

I am convinced that this will increase the interest in the submission.

Author Response

Dear reviewer,
thank you for your work and recommendations.

In the abstract please explain the abbreviation: TRIS-NPG (as in line 57).
We replace the abbreviation by the full name.

Line 58: too many references cited in one parenthesis.

Line 66: reference [24] should be cited earlier, in line 58.

We reduced the number of literature references and rearranged them.

Figure 1 caption: remove: “taken from”
We follow your recommendation.

Lines 122, 139, 149: remove power by unit.
We follow your recommendation.

Figure 2: I suggest describing both schemes as a) and b)
We follow your recommendation.

Table 1: add a row with column names: name, symbol, unit, value.
We follow your recommendation.

References: format according to the journal’s guidelines. Most of the cited papers have been published quite a long time ago. ……..

The number of literary figures has been reduced according to your recommendation. The reference to the already not nwerst publications is necessary because it concerns the theoretical approaches to understanding peritectic layered formation or the first experimental proof of the existence of peritectic layered formation.

We also thank you for the recommended publications. Since further publications are planned, we will find a corresponding mention in the context of the new publication.

Best regards,
Johann P. Mogeritsch

Reviewer 2 Report

The used method is rather pertinent, the obtained results are very interesting, the associated discussions are convaincing. To lead a better understanding, some details should added:

in nearly all figures (fig.3, 4, 5, 6, 7, 8, 9, 10, 12 and 13), it is necessary to add a scale in each photo to show the dimension of the observed microstructure; in discussion part, it is extremely important to cite previous works to support the theoritical or experimental observations;

Author Response

Dear reviewer,
thank you for your work and recommendations. We put in all from you listed figures in each left image the required length scale. Additionally, in the discussion, we remark the connection to already published papers.
Best regards,
Johann P. Mogeritsch